# Biological Redox Impact of Tocopherol Isomers Is Mediated by Fast Cytosolic Calcium Increases in Living Caco-2 Cells

**DOI:** 10.3390/antiox9020155

**Published:** 2020-02-14

**Authors:** Miltha Hidalgo, Vania Rodríguez, Christine Kreindl, Omar Porras

**Affiliations:** Laboratory for Research in Funtional Nutrition, Instituto de Nutrición y Tecnología de los Alimentos, Universidad de Chile, Av. El Líbano 5524, Macul, Santiago 7830490, Chile; miltha.hidalgo@inta.uchile.cl (M.H.); dravaniarodriguez@gmail.com (V.R.); ntakreindl@gmail.com (C.K.)

**Keywords:** tocopherol, HyPer, Caco-2, calcium imaging, redox

## Abstract

Most of the biological impacts of Vitamin E, including the redox effects, have been raised from studies with α-tocopherol only, despite the fact that tocopherol-containing foods carry mixed tocopherol isomers. Here, we investigated the cellular mechanisms involved in the immediate antioxidant responses evoked by α-, γ- and δ-tocopherol in Caco-2 cells. In order to track the cytosolic redox impact, we performed imaging on cells expressing HyPer, a fluorescent redox biosensor, while cytosolic calcium fluctuations were monitored by means of Fura-2 dye and imaging. With this approach, we could observe fast cellular responses evoked by the addition of α-, γ- and δ-tocopherol at concentrations as low as 2.5 μM. Each isomer induced rapid and consistent increases in cytosolic calcium with fast kinetics, which were affected by chelation of extracellular Ca^2+^, suggesting that tocopherols promoted a calcium entry upon the contact with the plasma membrane. In terms of redox effects, δ-tocopherol was the only isomer that evoked a significant change in the HyPer signal at 5 μM. By mimicking Ca^2+^ entry with ionomycin and monensin, a decline in the HyPer signal was induced as well. Finally, by silencing calcium with 1,2-bis(o-aminophenoxy)ethane-N,N,N′,N′-tetraacetic acid (BAPTA), an intracellular Ca^2+^ chelator, none of the isomers were able to induce redox changes. Altogether, our data indicate that an elevation in cytoplasmic Ca^2+^ is necessary for the development of a tocopherol-induced antioxidant impact on the cytoplasm of Caco-2 cells reported by HyPer biosensor.

## 1. Introduction

Vitamin E is a term that groups tocopherols and tocotrienols. However, the concept has evolved to refer mainly to α-tocopherol, likely due to its high relative abundance in plasma and blood cells [1]. This notion is changing now; the pharmacokinetics of the β, γ and δ isomers in the plasma of healthy individuals who were subjected to a single dose of a preparation enriched with one of the tocopherol isomers indicated that all the isomers reached circulation 8 h after intake of the respective isomer [2]. Other studies performed in healthy individuals [3,4], as well as in patients diagnosed with diabetes mellitus type 2 [5] who followed a daily regimen of a realistic mixture of tocopherols for weeks, showed significant increases in the concentration of α- and γ-tocopherol in circulating blood cells and serum compared to a placebo group.

On the other hand, the relative abundance of tocopherol isomers in food is quite diverse. Using electroanalytical techniques, Robledo et al. compared the content of tocopherol isomers in edible oils such as corn, soybean and canola, among others. Between them, γ-tocopherol represented more than 65% of the total Vitamin E content [6]. This pattern of abundance for non-α-tocopherols is also evident in 29 species of beans, in which γ-tocopherol and δ-tocopherol were the most abundant isomers, with only a discrete amount of α-tocopherol [7]. Interestingly, δ-tocopherol has also been detected in human milk (2.8 μM) [8,9,10]. Therefore, it is clear that early in life during breastfeeding or from the diet in adulthood, we are exposed not only to α-tocopherol but to all the isomers.

The antioxidant properties of tocopherol isomers determined by chemical-based assays have been well known since 1996 [11]. Particularly, the first encounter between tocopherols and the intestinal epithelial barrier is relevant, as was tackled by Elisia et al. using two cellular models of intestinal epithelium exposed for 24 h to α-, γ- and δ-tocopherol. The results indicated that γ- and δ-tocopherol were more efficient in protecting oxidation of plasma membrane in Caco-2 cells induced by a chemical reactive oxygen species (ROS) generator [9]. However, acute biological effects for the isomers of tocopherol are still unknown.

In this work, we followed the cytosolic Ca^2+^ and redox fluctuations induced by α-, γ- and δ-tocopherol isomers and α-tocotrienol in living Caco-2 cells. Both phenomena were tracked with a temporal resolution of seconds in order to gain information about the first cellular components affected by the tocopherol/tocotrienol arrival to the plasma membrane and if they participate in the redox properties of these compounds. Our findings indicate that upon exposure to isomers at concentrations as low as 2.5 μM, cytosolic Ca^2+^ levels experienced a fast increase, showing different kinetic patterns among the isomers. By chelating the extracellular Ca^2+^, the cytoplasmic Ca^2+^ elevations induced by all isomers were suppressed, indicating that the calcium response was dependent on Ca^2+^ entry. Similarly, differential redox responses were induced upon isomer exposure in HyPer-expressing Caco-2 cells with higher impact of δ-tocopherol at low concentrations than any of the other isomers. Finally, by mimicking an elevation in cytosolic Ca^2+^ with ionophores and avoiding Ca^2+^ responses with an intracellular calcium chelator, we unveiled a fast connection between calcium signalling and redox adjustment in living Caco-2 cells. In summary, our data show that the immediate antioxidant effect of tocopherol isomers is mediated by calcium signalling, which in turn is dependent on the opening of Ca^2+^ permeability at the plasma membrane of living Caco-2 cells.

## 2. Material and Methods

Reagents and salts for buffer preparation were obtained from Sigma-Aldrich (St. Louis, MO, USA). Media culture and supplements were purchased from Life Technologies (Carlsbad, NY, USA). Ionophores like ionomycin and monensin were acquired from Tocris (Bristol, UK). Fura-2 AM and SBFI-AM dyes, along with pluronic acid, were acquired from Thermo Fisher Scientific (Waltham, MA, USA). Tocopherols (α, γ, δ) were purchased from Sigma-Aldrich (St. louis, MO, USA), α-tocotrienol was purchased from Cayman Chemicals Company (Ann Harbor, MI, USA). Stock solutions were prepared in absolute ethanol at 50 mM. The 30% hydrogen peroxide solution and absolute ethanol used in this work were from Merck (Darmstadt, Germany).

### 2.1. Cell Culture

Caco-2 cells (HTB-37^TM^, colorectal adenocarcinoma) were purchased from the American Type Culture Collection (Manassas, VI, USA) and cultured in Minimum Essential Medium supplemented with 10% fetal bovine serum. In general terms, cells were maintained under a humidified atmosphere with 5% CO_2_/air, and culture medium was renewed every 2 or 3 days. When 70–80% confluence was reached, cultures were expanded to other plaques or seeded on glass coverslips (Marienfeld, Germany) to be further imaged. Cells were used at passages 20–40.

### 2.2. Ca^2+^ and Na^+^ Imaging

For imaging experiments, the coverslips were mounted in an open recording chamber, and media were replaced by Krebs Ringer HEPES (KRH) buffer (in mM: 140 NaCl, 4.7 KCl, 20 HEPES, 1.25 MgSO_4_, 1.25 CaCl_2_; pH 7.4) supplemented with 5 mM glucose (KRH-glc). Dyes were prepared in 0.01% pluronic acid in DMSO solution; 5 μM of the selected dye was used to load the cells for 30 min at room temperature in KRH-glc. After that time, cells were rinsed three times with KRH-glc and left for another 20 min to allow complete de-esterification of the acetoxymethyl ester (AM) motif from the remanent intracellular dye.

Imaging took place in an inverted Nikon Ti microscope (Tokio, Japan) equipped with 40× oil objectives (numerical aperture, N.A. 1.3). A xenon lamp was coupled to the monochromator device (Cairn Research Ltd., Faversham, UK), which allowed dual excitation at 340 and 380 nm for Fura-2 and SBFI, respectively. Emission over 520 nm was collected by a long-pass filter, fluorescence was digitalized by a cooled charge-coupled device (CCD) camera (Hamamatsu ORCA 03, Hamamatsu, Japan) and image analysis was performed using free micromanager software (San Francisco, CA, USA) [12]. Data sampling was acquired every 10 s and expressed as 340/380 ratio.

### 2.3. HyPer Biosensor Imaging

The HyPer biosensor was introduced into the cytoplasm of Caco-2 cells by infection with adenovirus at 1:100 dilution. Adeno-particle production has been explained in detail by our group [13]. Briefly, cyto-HyPer cDNA (Evrogen, Moscow, Russia) was subcloned into the commercial adenoviral vector pAdEasy-RFP using conventional molecular biology techniques, and homologous recombination was done using BJ5183 cell transformation. AdHek cells were used to expand and replicate the viral particles. After 21 days of infection, cells were harvested and subjected to three freeze-thaw cycles followed by centrifugation to remove cellular debris. The resulting supernatant (2 mL) was stored at −80 °C or used to infect target cells.

Two days after infection, HyPer was homogeneously expressed in the cytosol with no signs of cytotoxicity, and cells were healthy and full-adhered to the glass surface, ready for imaging. As described for calcium imaging, coverslips containing HyPer-expressing Caco-2 cells were mounted in an open recording chamber with KRH-glc buffer. The HyPer biosensor was excited at 420 and 490 nm, whereas the emitted light was collected with a long-pass filter over 520 nm. For every recording, an initial lapse of 20 min was done to ensure a reliable and stable baseline, which functions as the basal value.

### 2.4. Statistical Analysis

Throughout the manuscript, data are expressed as mean ± standard error. Paired Student’s *t*-tests were executed only for comparisons of before-after experimental designs, whereas comparisons of multiple groups or repeated measurements were both evaluated by ANOVA with Bonferroni post hoc analysis for parametric data or Dunn’s method for nonparametric data. A *p*-value less than 0.05 was considered statistically significant. In order to classify cells that elicited a positive calcium response to tocopherol exposure, which we called responding cells, we defined a threshold of 3 standard deviations (SD) over baseline value to consider an increase in the cytosolic calcium level; this analysis was performed by comparing mean values and SD obtained from 3-min bins before tocopherol exposure and two sequential bins after tocopherol exposures (3 or 6 min). A similar analysis was performed with HyPer data as well. Ca^2+^, Na^+^ and redox cellular responses elicited by tocopherol/tocotrienol isomers took place on cells with the same passage and the same day. 

## 3. Results

### 3.1. Tocopherol Isomers Evoked Rapid Increases in Cytosolic Ca^2+^ with Different Patterns in Caco-2 Cells

By loading Caco-2 cells with Fura-2, a ratiometric Ca^2+^ indicator, we detected rapid increases in Ca^2+^ upon exposure to α-, γ- and δ-tocopherol at concentrations as low as 2.5 μM (Appendix A). As shown in Figure 1, cell population responded heterogeneously and differently to the addition of 5 μM of these three isomers. Right after the application of α- and γ-tocopherol, few cells showed rapid increases in cytosolic Ca^2+^; however, these immediate responses were not observed in any of the Caco-2 cells exposed to δ-tocopherol, which induced Ca^2+^ increases that developed gradually during the first 3 min of exposure. According to the different temporal pattern of tocopherol induced-Ca^2+^ increases, 67% (52 out of 78 cells) of Caco-2 cells were responsive to α-tocopherol during the first 3 min of exposure, which augmented to 86% in the next 3 min of exposure. Under the same analysis, the percentage of cells responding to γ-tocopherol was more stable; 60% and 52% of 79 imaged cells were quantified in the first 3 min and the next 3 min of exposure, respectively. δ-tocopherol, on the other hand, showed 68% of cells responding during the first 3 min, which evolved to 97% for the next 3 min of recording (31 single-cell recordings). From this real-time Ca^2+^ imaging, we presented the differential acute effects on cytosolic Ca^2+^ evoked by tocopherol isomers, in which δ-tocopherol-induced Ca^2+^ increases developed more gradually than those induced by the other isomers. In addition to tocopherols, we also evaluated the capacity of α-tocotrienol to induce cytosolic Ca^2+^ increases. Exposure to 5 μM α-tocotrienol did not evoke significant changes in the cytoplasmic Ca^2+^ levels; however, at 50 μM, some cells (4 out of 17) experienced Ca^2+^ augments (Appendix A). Data analysis of the calcium responses induced by δ-tocopherol after 6 min of exposure indicated a sustained increase in the cytoplasmic Ca^2+^, a phenomenon that was not observed with the other isomers. In terms of the magnitude, δ-tocopherol induced bigger Ca^2+^ responses in most of the imaged cells compared to the treatment with the other isomers (Appendix A).

### 3.2. Removal of Extracellular Ca^2+^ Abolishes the Cytosolic Ca^2+^ Increases Induced by All Tocopherol Isomers

In order to clarify the source of Ca^2+^ that contributes to the rapid cytosolic Ca^2+^ increase evoked by the application of tocopherol isomers, we tested if 50 μM of each isomer was able to evoke a cytosolic Ca^2+^ increase in cells maintained in a KRH buffer without Ca^2+^ and supplemented with 5 mM Ethylene glycol-bis(β-aminoethyl ether)-N,N,N′,N′-tetraacetic acid (EGTA), a calcium chelator. Under Ca^2+^ free medium, the number of tocopherol-responding cells and the magnitude of calcium responses diminished significantly. It is noteworthy to point out that in the absence of extracellular Ca^2+^, the immediate peak evoked either by α- or γ-tocopherol disappeared, suggesting that this fast component of the calcium response has an extracellular source. On the other hand, Ca^2+^ responses evoked by δ-tocopherol, which lacked this early peak, elicited a slow development, an observation that was noticeable even in the absence of extracellular Ca^2+^, indicating that δ-tocopherol may trigger a calcium mobilization from intracellular sources (Figure 2).

Until now, Caco-2 cells seem to respond differentially to tocopherol isomers in terms of Ca^2+^ patterns and more specifically in the rapid component visible for α- and γ-tocopherol, which presents a notable dependency on extracellular Ca^2+^, suggesting different mechanisms of interaction or recognition at the cellular surface, likely promoting the opening of calcium permeability at the plasma membrane. On the other hand, δ-tocopherol-induced calcium responses developed gradually and lacked the fast component observed in α- and γ-induced stimulations, suggesting that this phenomenon was more related to calcium mobilization from intracellular sources.

### 3.3. Only δ-Tocopherol Evokes an Acute Redox Effect on Caco-2 Cells at Low Concentration

Next, we evaluated the acute impact of tocopherol isomers on cellular redox at steady state in living Caco-2 cells by means of the HyPer biosensor. This biosensor carries a pair of cysteine residues that only in the presence of H_2_O_2_ form a disulfide bond, which can be restored by the antioxidant capacity of the cytosolic environment. In Figure 3, we compare the effects of exposure to 5 μM α-, γ- and δ-tocopherols on the HyPer signal. Only δ-tocopherol was effective in inducing a significant diminishment in the biosensor signal; application of α-tocopherol provoked a slight decline in the signal, which did not become significant during the temporal window that we evaluated. γ-tocopherol, on the other hand, did not affect the HyPer signal during the whole recording period. Although HyPer imaging was carried out every 20 s, it is evident that the redox effect induced by δ-tocopherol is rapid and develops gradually during the 10 min of recording until the signal reaches a plateau. These results indicate that at 5 μM, δ-tocopherol is the only isomer capable of inducing a rapid change in the biosensor signal. This observation is also valid for 5 μM α-tocotrienol, which did not affect the baseline upon exposure (Appendix A). By increasing the concentration used to 50 μM, all the tocopherol isomers, but not α-tocotrienol, induced a similar response in Caco-2 cells expressing HyPer (Appendix A).

### 3.4. Massive Calcium Entry Using Ionophores Induced a Fast Redox Response 

At this point, we have shown two acute effects induced by tocopherol isomers on Caco-2 cells: rapid cytosolic Ca^2+^ increases together with fast changes in the HyPer signal. However, the question of whether this rapid calcium increase relates to the redox response is still pending. Thus, to address this issue, we first provoked a massive Ca^2+^ entry by applying a well-known calcium ionophore, ionomycin. Figure 4 shows that the cytosolic calcium of Caco-2 cells increases rapidly after the addition of 2 μM ionomycin; the same stimulus evokes a diminishment in the ratiometric signal of HyPer, indicating that direct manipulation of cytosolic Ca^2+^ levels, independent of the tocopherol isomer, is enough to trigger a redox change in Caco-2 cells. Complementary to this approach, we used monensin, an Na^+^/H^+^ antiporter that induces a fast cytosolic Na^+^ load together with a secondary calcium increase in the cytosolic compartment, probably by the activation of Na^+^/Ca^2+^ exchangers (Figure 5). As with ionomycin, HyPer signal experienced a decrease in the presence of monensin, a phenomenon that was not observed in HyPer-expressing Caco-2 cells loaded with BAPTA (Figure 5A,B), an intracellular calcium chelator that does not interfere with the Na^+^ entry, as can be observed in Figure 5C. These experiments suggest that increments in cytoplasmic Na^+^ are not connected with redox changes, while Ca^2+^ augmentation triggers a change in the antioxidant environment that is reported by the HyPer biosensor.

### 3.5. Acute Redox Changes Induced by Isomers of Tocopherol Are Dependent on Intracellular Calcium Increases

Finally, to establish if Ca^2+^ increment induced by tocopherol isomers were effectively mediating changes in the antioxidant capacity of the cytoplasm, which are reported by HyPer as a diminishment in the ratio signal, we treated the cells with BAPTA and then applied 50 μM of each tocopherol isomer, a concentration that induces redox changes for the three isomers evaluated as can be observed in Appendix A. None of the evaluated isomers were able to induce a diminishment in the signal of HyPer of BAPTA-treated Caco-2 cells, confirming that elevations in cytoplasmic calcium are required to induce acute redox responses (Figure 6). This finding expands the level of interaction of tocopherols beyond their potential as ROS scavengers, by adding calcium signalling as part of its repertoire to induce redox responses in intact mammalian cells.

## 4. Discussion

Intestinal cells represent one of the first host barriers that interact with dietary compounds and therefore are exposed to the potential antioxidant effect that some molecules possess, like tocopherol isomers, which, according to their relative abundance in food, are mainly represented by α- and γ-tocopherols followed by δ-tocopherol. Here, we monitored the impact of short-term exposure of these three isomers on the cytoplasmic calcium level and redox status in living Caco-2 cells with a time resolution of seconds to minutes in order to visualize the first contact of these compounds with a cellular model for intestinal wall. Our experimental approach allowed us to record different kinetic patterns of cytoplasmic Ca^2+^ increases depending on the isomer used, as well as the contribution from external Ca^2+^ to these responses. In terms of redox effect, δ-tocopherol was the only isomer capable of modifying the HyPer signal at 5 μM. Further experiments were addressed to establish a mechanistic connection between these two phenomena observed in the cytoplasm. By inducing a massive entry of extracellular Ca^2+^ with both ionomycin and monensin, we evoked a redox change like those observed for tocopherol exposures. However, based on the intrinsic properties of these ionophores, we could discard the role of cytoplasmic Na^+^ load in the redox response of Caco-2 cells. Finally, intracellular calcium chelation was effective in avoiding the redox change induced by tocopherol isomers. Our data together shed light on a connection between rapid Ca^2+^ and redox responses in living Caco-2 cells, a phenomenon that helps to understand how redox homeostasis is finely tuned with some cellular signalling triggered by functional dietary compounds.

All three tocopherol isomers evaluated induced calcium responses within seconds to minutes, albeit with some variability through the recordings. In Caco-2 cells, tocopherol uptake is mainly mediated by cholesterol transport carriers such as Scavenger Receptors class B type I (SR-BI) [14] and Niemann-Pick C1-Like1 (NPC1-L1) [15]. SR-BI is a receptor widely expressed in several types of cells that not only facilitates cholesterol transport but also mediates cellular signalling triggered by HDL, for example. Regarding Ca^2+^ signalling evoked by HDL, Lee at al. demonstrated that a concomitant activation of Sphingosine-1-Phosphate Receptor (S1PR) was necessary to mobilize cytosolic calcium in primary rat aortic smooth muscle and HEK293 cells [16]. Interestingly, the association between SR-BI and S1PR induced by sphingosine-1-phosphate provoked a remarkable Ca^2+^ increase that developed rapidly and transiently. However, the authors did not explore the contribution of extracellular calcium, assuming that most of the calcium effect was due to protein G-coupled receptor signalling. Our experiments indicate that the fast cytoplasmic calcium responses induced by α- and γ-tocopherols (50 μM) were abolished by the removal of extracellular Ca^2+^, suggesting an interaction of these isomers with other entities capable of permeating this divalent ion through the plasma membrane of Caco-2 cells. TRPC5, a member of the transient receptor potential (TRP) family, is a candidate for connecting calcium signalling with the extracellular presence of antioxidant compounds, since this channel presents an extracellular pair of cysteines able to form a disulfide bond; this structural feature works as a redox switch that controls the subcellular traffic, stabilization and activity of this channel [17]. DTT and thiorredoxin, both added extracellularly, generated an increase of nonselective cationic currents mediated by this channel in HEK293 cells and on the endogenous channel present in the joint tissue obtained from patients diagnosed with rheumatoid arthritis [18]. In addition to synoviocytes, TRPC5 has been also detected in cellular models like IC-6 (rat) and Caco-2 (human) at the same passage that was used in this work, and therefore it is a suitable molecular candidate to connect extracellular redox tone with calcium-dependent processes [19]. All the tocopherol isomers tested here have the capacity to mobilize cytoplasmic Ca^2+^ in Caco-2 cells, a general phenomenon that has the potential to promote the intestinal barrier function by stimulating mucin production, as demonstrated in HT-29 cell line [20], or stimulating mucin granules secretion in human goblet cells, a cellular processes mediated by the activation of TRPM5, a Ca^2+^-activated cationic channel [21]. Increases in cytosolic Ca^2+^ might also improve the thickness and stagnancy of secreted mucus by participating in the secretion of the calcium-activated chloride channel regulator I, a very abundant metalloprotease in intestinal cells. Once at the extracellular space, this metalloprotease is able to process the inner mucus layer, modifying its physicochemical properties [22]. Our experiments performed without extracellular Ca^2+^ have unveiled an interaction between tocopherols and molecular entities at the plasma membrane that conduct calcium responses, which in turn affect the redox homeostasis of intestinal cells.

Regarding the acute redox effects elicited by tocopherol molecules recorded in this work, we consider the plasma membrane as the first cellular structure where tocopherol must interact. Ultrastructural studies on artificial bilayers indicate that the hydroxyl group located at the chromanol ring of α-tocopherol molecules faces the aqueous interface in both extra- and intracellular compartments, suggesting that tocopherols can exert their antioxidant feature not only restricted to the central lipid environment but also connecting the soluble components from the extra- and intracellular environment of living cells [23]. However, in our cellular model, cytoplasmic redox changes were evoked by ionophores without intrinsic antioxidant properties, which provoke a massive Ca^2+^ entry in terms of seconds, suggesting that the disulfide bond reduction observed in the biosensor occurred by a Ca^2+^-dependent cytoplasmic mechanism and not by a direct antioxidant action of tocopherols. Such calcium-coupled redox events have been observed in HEK293 cells by simultaneous tracking of cytoplasmic calcium and H_2_O_2_ after a short exposure to thapsigargin (2 min). This treatment induced a transient cytoplasmic calcium elevation that decreased gradually with time. HyPer signal, on the other hand, exhibited an initial small increase, to later, in approximately 10 min, stabilize at a new level with lower values than those observed at the baseline. These results indicate that the redox tone in the cytoplasm of HEK293 reaches a more reduced level, triggered initially by the thapsigargin-evoked Ca^2+^ increase [24]. In primary hippocampal neurons, HyPer signal recovery is also accelerated by the neurotransmitter glutamate, an effect that is mediated by calcium influx and stimulation of mitochondrial functions [25]. Here, we have not explored metabolic activation of Caco-2 cells which might be involved in the global shift to a more reducing cytoplasmic environment.

We have exposed differential acute effects of tocopherol isomers on the cytoplasmic redox status, a phenomenon that requires the mobilization of calcium, suggesting that tocopherols, beyond their redox properties as ROS scavengers, have a redox impact by interacting with the cellular machinery of human intestinal cells. In other cellular models, such tocopherol as tocotrienol isomers has been studied regarding its capacity to protect cells from an oxidative challenge. For instance, 24 h preincubation with α-tocopherol or α-tocotrienol showed no differences in avoiding ROS generation in SH-SH5Y cells subjected to 1 μM H_2_O_2_ for 40 min [26]. In cerebellar granular cells, on the other hand, 5 μM α-, γ-tocotrienol showed axonal and dendritic protection from structural damage induced by 0.5 μM H_2_O_2_ (24 h) with some differences in the level of lipid peroxidation induced by H_2_O_2_. Certainly, the longer the exposure with tocopherol/tocotrienol, the more chances the compounds have to interact with cell components. In osteocyte-like cells, 2 h of treatment with δ-tocotrienol (25 μM) was effective to protect cells against death induced by 3 h of *tert*-butyl hydroperoxide (250 μM), a protection that was sensitive to the pharmacological inhibition of Nrf2 as well as PI3K signalling [27], supporting the idea that antioxidant properties of tocopherol/tocotrienol are not limited by their ROS scavenging capacity, but also to functional interaction with cellular components.

The tocopherol concentrations used in this work were between 2.5 to 50 μM, a range frequently found in plasma of healthy subjects [28] or subjected to supplementation [29,30]. However, local tocopherol concentrations that colonic cells face are more relevant. In the case of infants, tocopherol isomers have been detected in faecal samples by HPLC from two groups, before (5 months old) and after receiving complementary food (6–12 months); such determinations allow an estimation of 20 to 160 μM of α-tocopherol [31]. Data obtained from right colon biopsies performed in adult subjects with normal colonic mucosa, as reported by Nair et al., indicate that α- and γ-tocopherols were found at concentrations of 17 and 6 μg/g, respectively [32], indicating a luminal concentration of tocopherols in the order of tens to hundreds of nanomolar. However, the primary interaction of dietary compounds with molecular entities at the plasma membrane of intestinal cells deserves to be further investigated to establish molecular mechanisms of healthy diet effects based on physiological impacts.

## 5. Conclusions

Tocopherol isomers evoked fast calcium responses in Caco-2 cells with different kinetics.δ-tocopherol was the only isomer capable of inducing a redox change in HyPer biosensor at 5 µM.An increase in cytosolic Ca^2+^, but not in Na^+^, was necessary and sufficient to induce a reduction in the HyPer biosensor for all the isomers evaluated.A massive cytosolic Ca^2+^ increase induced by both ionophores, ionomycin and monensin, was enough to generate a redox change in HyPer.

## Figures and Tables

**Figure 1 antioxidants-09-00155-f001:**
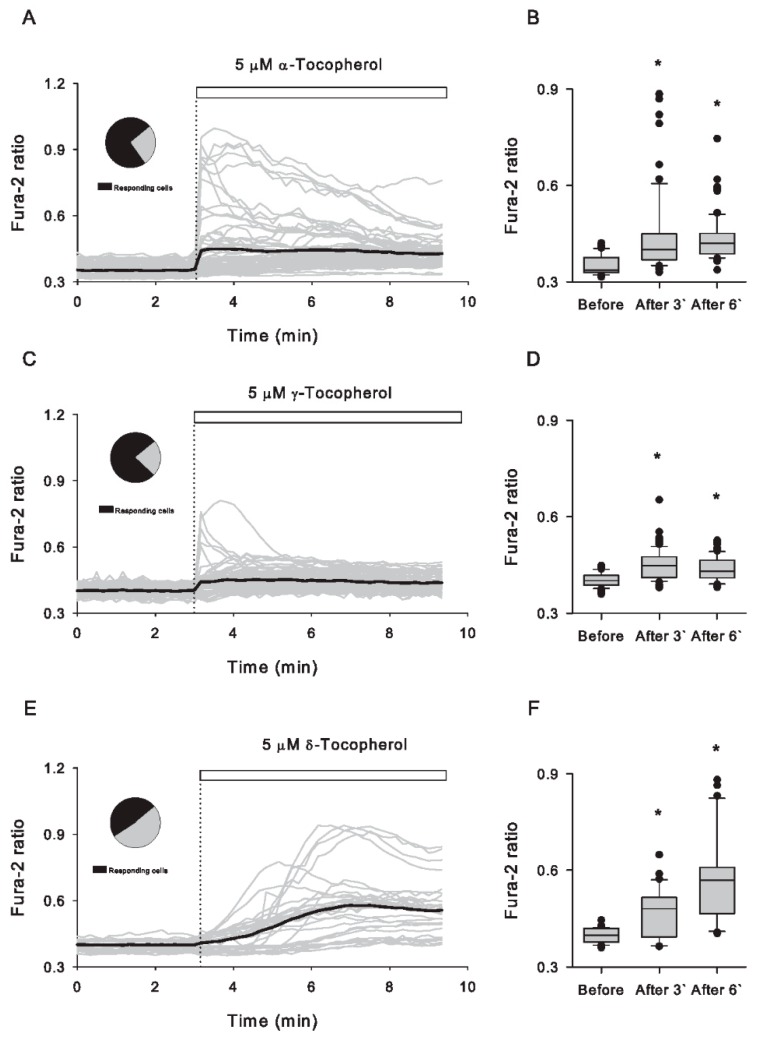
α-, γ- and δ-tocopherol isomers induce acute cytosolic Ca^2+^ increases with different patterns in Caco-2 cells. (**A**) Time course of Fura-2 fluorescence ratio in Caco-2 cells; grey traces in the graph correspond to 78 single-cell recordings from three independent experiments, whereas the black line depicts their average. The moment of 5 μM α-tocopherol addition is indicated by a dotted line and by the white bar on the plot. The pie chart illustrates the proportion of responding cells in black. (**B**) Quantification of fluorescence ratio values taken from a 3 min bin before and after the addition of α-tocopherol (3 and 6 min); the line into the boxes corresponds to the median, black symbols (●) correspond to potential outliers. (**C**) Same as in (**A**), but in this graph the effect of 5 μM γ-tocopherol is shown by 79 single-cell recordings from four independent experiments in grey traces, whereas the black line represents the average of these responses. The pie chart illustrates the proportion of responding cells in black. (**D**) Comparison of Fura-2 ratio values obtained as described for (**B**). (**E**) Same as described in (**A**) and (**C**), but this graph shows the effect of 5 μM δ-tocopherol obtained from 31 single-cell recordings of five independent experiments. The pie chart illustrates the proportion of responding cells in black. (**F**) Analysis of fluorescence ratio values obtained before and after application of this isomer as described in (**B**) and (**D**). Asterisks (*) indicate significant differences obtained by paired Student’s *t*-test.

**Figure 2 antioxidants-09-00155-f002:**
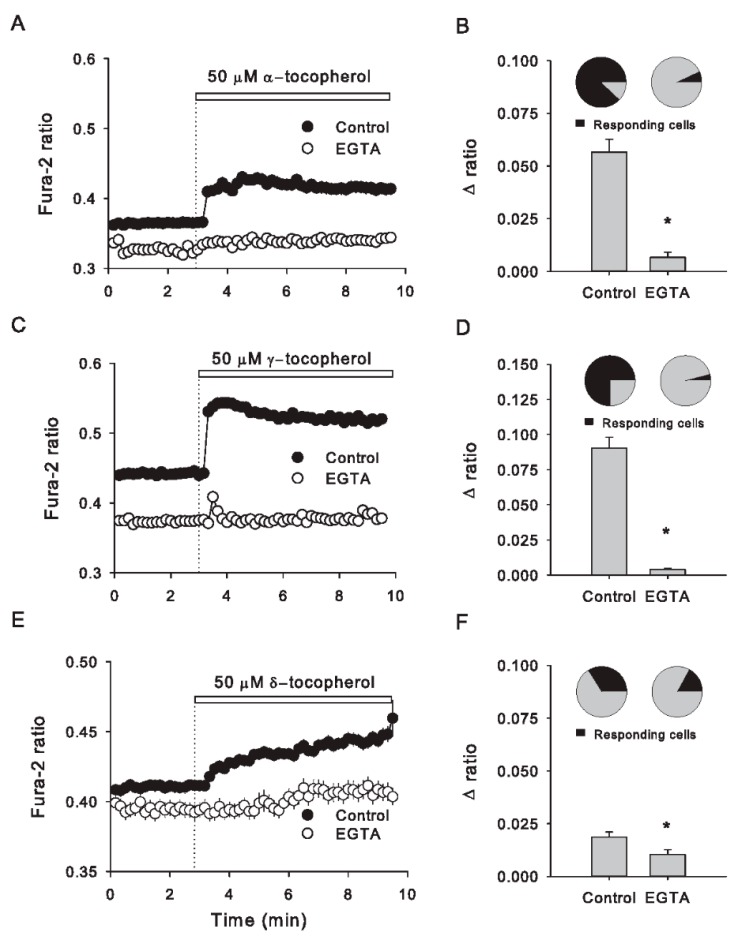
Removal of extracellular Ca^2+^ abolishes intracellular Ca^2+^ increases induced by all tocopherol isomers in Caco-2 cells. (**A**) Time course of Fura-2 fluorescence ratio in Caco-2 cells in the presence of an extracellular buffer without Ca^2+^ and supplemented with 5 mM EGTA (EGTA, open circles) or in a normal condition with Ca^2+^ (control, filled circles); 50 μM α-tocopherol was added at the time indicated by the dotted line and the white bar on the plot. Data correspond to the average ± SE of 41 control cells from three independent experiments and 146 EGTA-treated cells from six independent experiments. As described in (**A**), the plot in (**C**) shows the effect induced by 50 μM γ-tocopherol. Data correspond to the average ± SE of 44 control cells from three independent experiments and 109 EGTA-treated cells from four independent experiments. (**E**) The effect of δ-tocopherol is shown. Data correspond to the average ± SE of 63 control cells from four independent experiments and 110 EGTA-treated cells from seven independent experiments. In (**B**,**D**,**F**), cytosolic calcium increases were quantified as the subtraction between average values of Fura-2 ratio obtained 3 min before and after those of α-, γ and δ-tocopherol isomers addition in control and EGTA-treated cells, respectively. Above each bar, a pie chart represents the proportion of responding cells for each experimental condition. The asterisks mean statistical differences obtained by nonpaired t-Student test.

**Figure 3 antioxidants-09-00155-f003:**
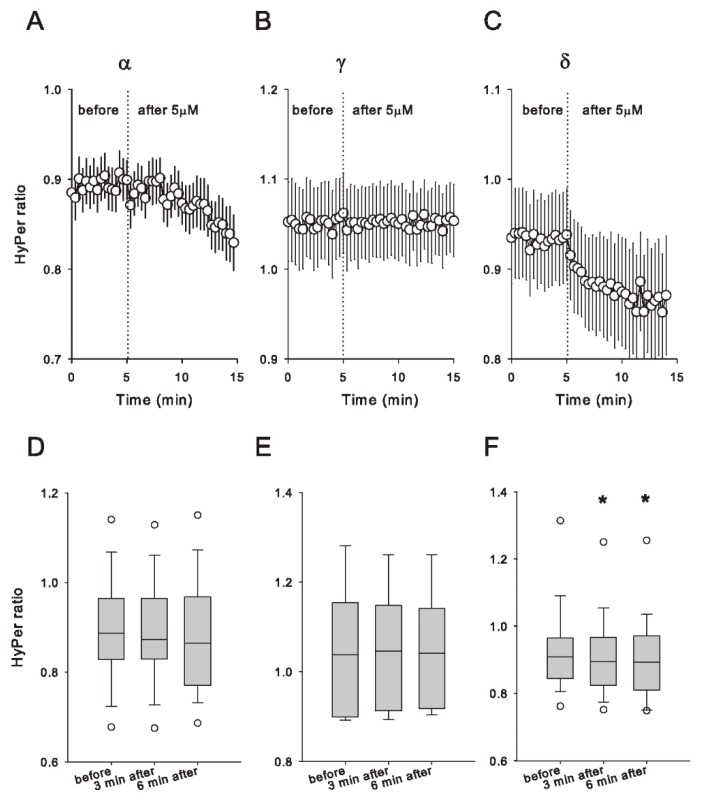
Acute effects of tocopherol isomers on HyPer baseline signal in Caco-2 cells. Caco-2 cells were transduced with the HyPer biosensor. Two days after, HyPer imaging took place. Five micromoles of α (**A**), γ (**B**) and δ (**C**) isomers was applied as indicated by dotted lines in the graphs. Data correspond to the mean ± SE of at least 11 cells from three independent experiments. For each single-cell recording, HyPer ratio values were obtained before and after isomer addition (3 and 6 min) of α- (**D**), γ- (**E**) and δ- (**F**) tocopherol. The lines in the middle of the boxes correspond to the median of the data group. Open circles (◦) correspond to potential outliers. Asterisks (*) mean significant differences between groups according to RM-ANOVA against the control group (Dunn’s method).

**Figure 4 antioxidants-09-00155-f004:**
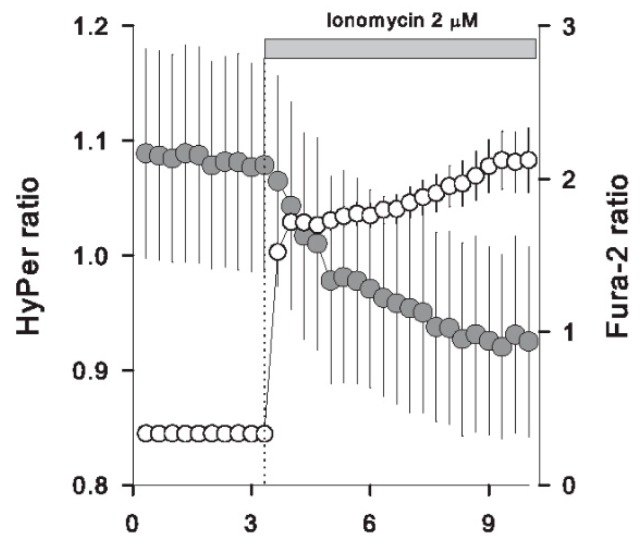
Ionomycin induces a diminishment in the HyPer signal. Caco-2 cells loaded with Fura-2 dye (open circles, 21 cells from three independent experiments) and another set of Caco-2 cells expressing HyPer (grey circles, 11 cells from three independent experiments) were exposed to 2 μM ionomycin at the time indicated by the dotted line and grey bar on the plot. Data correspond to the average ± SE.

**Figure 5 antioxidants-09-00155-f005:**
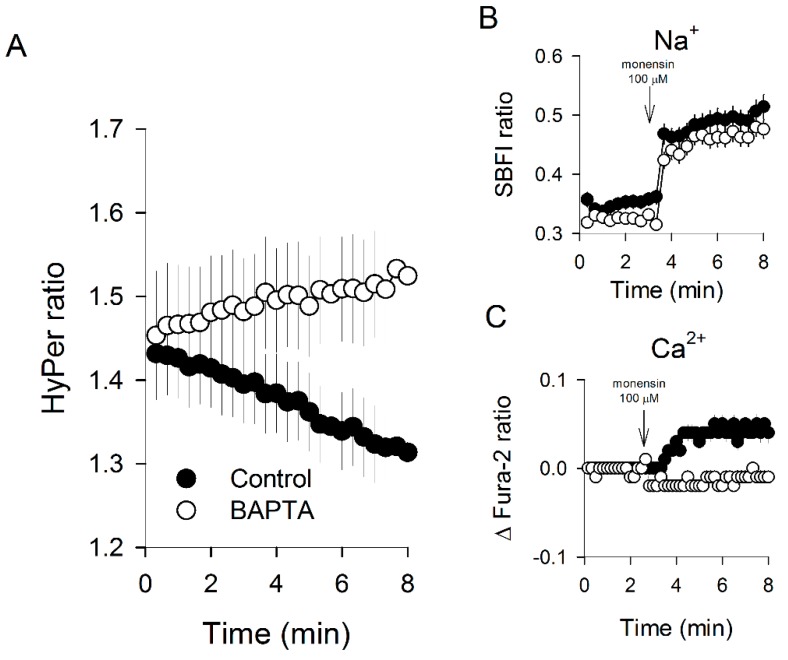
Cytosolic calcium increment is necessary to observe a monensin-induced HyPer signal decrease in Caco-2 cells. (**A**) HyPer signal obtained 2 min after addition of 100 μM monensin in control Caco-2 cells (filled circles) and in cells loaded with 75 μM BAPTA (open circles). Data are expressed as the average ± SE of 18 control cells and 15 BAPTA-loaded cells from three independent experiments. (**B**) Cytosolic sodium increment was measured in Caco-2 cells by means of SBFI imaging; data correspond to the average ± SE of 58 control cells from four experiments (filled circles) and 32 cells from three experiments treated with BAPTA (open circles). (**C**) Changes in the fluorescence ratio of Fura-2 were recorded for 61 control cells from four experiments (filled circles) and 35 cells loaded with BAPTA from three experiments (open circles). Data correspond to the average ± SE; the addition of 100 μM monensin in (**B**) and (**C**) is indicated by arrows on the plots.

**Figure 6 antioxidants-09-00155-f006:**
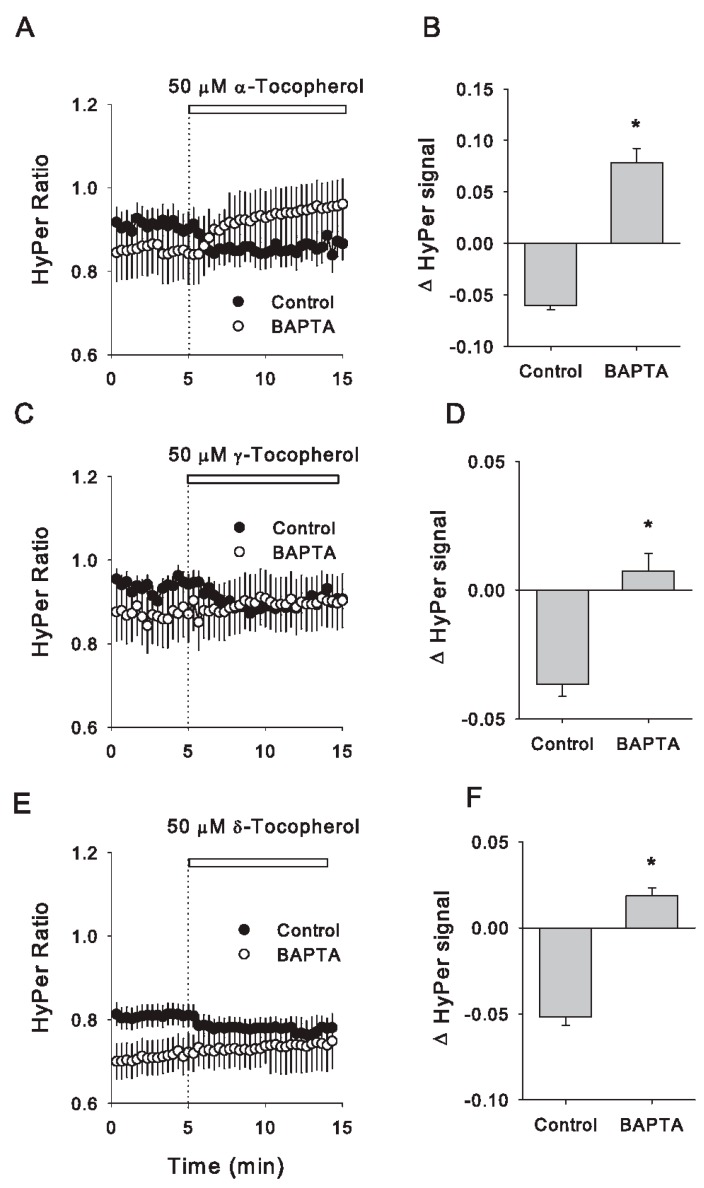
Intracellular calcium chelation abolishes a redox effect of tocopherol isomers on HyPer-expressing Caco-2 cells. Time courses of HyPer signal showing the effect of the three tocopherol isomers (50 μM) in nontreated cells (filled circles) and cells treated with 75 μM BAPTA (open circles). Isomer addition is shown by a dotted line and a white bar on each plot. (**A**), α-tocopherol was applied in 34 control cells from four experiments and 12 cells treated with BAPTA from three experiments. (**B**), changes in HyPer ratio induced by α-tocopherol exposure expressed as the difference between the signals obtained after 3 min of tocopherol addition and its respective baselines; this analysis was done for control and BAPTA-treated cells. As for (**A**,**B**), (**C**) shows the effect of adding of γ-tocopherol, whereas data analysis is depicted in (**D**). Data correspond to the average ± SE from 44 control cells from four experiments and 16 BAPTA-treated cells from three independent experiments. (**E**) The effect of δ-tocopherol is shown in 24 control cells from four experiments and 12 BAPTA-treated cells from three independent experiments. Data are expressed as the average ± SE. Data analysis is shown in (**F**). Asterisks (*) mean significant differences between control and BAPTA-treated groups (nonpaired t-Student test).

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
