# Peer review of "Biological Redox Impact of Tocopherol Isomers Is Mediated by Fast Cytosolic Calcium Increases in Living Caco-2 Cells"

_antioxidants, 2020, doi:10.3390/antiox9020155_

Round 1
Reviewer 1 Report
The study of Hidalgo et al is the resubmission of a former manuscript, which was also reviewed by the same reviewer.
The authors asked for more time to deal with the suggestions and to address them experimentally and decided to resubmit afterwards a revised version of their study. I appreciate the time and efforts that authors have done to address my concerns and in my opinion the paper has gained much more impact. As mentioned before I think this study is interesting for a broad readership and in the clear focus of the journal. I have only a few minor comments before this interesting and important study is suitable for publication.
- The authors have added some tocotrienol data and fully addressed this point
- Missing information in the material and method sections are added
- The discussion is more balanced now
- I would have appreciated if the authors would have confirmed their results in another cell line. However, the limitations are clearly addressed and even the headline states that these results are (only) obtained in Caco-2 cells.
- I partially agree to the points in respect to the virus-based transfection. The authors have stated that in the first days no cytotoxicity caused by the virus occurred and should therefore not influence the results. Please add this statement in material and method section.
- There is another paper out dealing with the impact of different tocopherols and tocotrienols in Alzheimer’s disease also in respect with ROS. Probably you could compare / discuss your results with this study. (PMID: 27801864)
Author Response
Dear reviewer,
I would have appreciated if the authors would have confirmed their results in another cell line. However, the limitations are clearly addressed and even the headline states that these results are (only) obtained in Caco-2 cells.
R: thanks for understand our point. We usually work with many cell types, most of our imaging data are related to redox impact with no Calcium and Sodium imaging as we put here in this manuscript. We are convinced that cells will bring different redox responses, with different mechanisms. Unfortunately, we haven run such sort of experiments in excitable cells, which I found an interesting issue without a doubt.
I partially agree to the points in respect to the virus-based transfection. The authors have stated that in the first days no cytotoxicity caused by the virus occurred and should therefore not influence the results. Please add this statement in material and method section.
R: a short phrase was added in that section (line 110-111)
There is another paper out dealing with the impact of different tocopherols and tocotrienols in Alzheimer’s disease also in respect with ROS. Probably you could compare / discuss your results with this study. (PMID: 27801864)
R: we have worked in a complete paragraph to deal with this suggestion. please find it in the discussion line 366-378).
Reviewer 2 Report
Please publish as is - the authors were good enough to adopt my suggestions
Author Response
Dear reviewer,
Thanks, we finished with an improved version of our ms after your suggestions.
OP
This manuscript is a resubmission of an earlier submission. The following is a list of the peer review reports and author responses from that submission.
Round 1
Reviewer 1 Report
The study of Hidalgo et al deals with the impact of vitamin E on cytosolic calcium increase and the redox potential. Importantly the authors point out that vitamin E consists of different tocopherols and distinguish between alpha, gamma and delta tocopherol in a consistent way. Key findings are: Each of the studied tocopherols increase cytosolic calcium level; an effect which can be attenuated by extracellular calcium chelators. Interestingly only delta tocopherol was able to alter the “redox status” measured by a HyPer assay. Experiments, increasing the intracellular calcium levels showed similar results on the redox status. An effect which could be silenced with intracellular calcium chelators.
The topic is in the clear focus of the journal and interesting for a broad readership, especially for scientists dealing with nutritional supplements, vitamin E, redox potential and calcium homeostasis. The findings are also important for several diseases e.g. several neurodegenerative diseases, where calcium or oxidative stress have an important impact. To my knowledge this is the first time where different vitamin E isoforms are investigated in respect to their calcium entry and their redox potential in this way.
Unfortunately, many important aspects were not addressed in this study and the underlying mechanisms remain unclear. Therefore, I cannot recommend to publish this study at this stage and I recommend the authors to put some more effort to clarify the remaining questions before this very important study should be published.
General remarks:
The authors use caco-2 cells. Caco-2 cells are derived from a colon carcinoma and widely used for in vitro uptake studies. The author correctly state that caco-2 cells are used as a cellular model for intestinal epithelium (line 66) and a previous study report that differentiated (!) caco-2 cells are influenced by different tocopherols in respect to their oxidative effect. However, for this study the rational for using these cells are not explained sufficiently and – in my opinion – using only (undifferentiated) caco-2 cells limit the biological relevance of their findings. I recommend that at least key experiments should be repeated in a neuronal or neuroblastoma cell line, which are widely used for calcium homeostasis and which would also emphasize the effect of vitamin E in brain and related disorders.
Especially tocotrienol is known to be antioxidative. Why do the authors limit their studies to the tocopherols? At least one tocopherol isoform should be compared to the corresponding tocotrienol to see if similar or even increased effect strengths are observed.
Major remarks:
- Actually, HyPer is a H2O2-sensitive fluorescent biosensor (for details see Insights into the HyPer biosensor as molecular tool for monitoring cellular antioxidant capacity by Hernandez et al). Moreover, an Adeno Virus system is used to transfect cells, which might influence the membrane as well. The authors should confirm their results by an additional more specific method, which has not these potential severe impact on the membrane and the metabolism of the cells.
- The authors should give mechanistical insights how the calcium homeostasis is influenced? It is completely unclear if any calcium channels are involved, which are involved etc.
- The conclusion should be more balanced. For example, the authors state that an elevation of cytoplasmatic calcium is necessary for development of a cytoplasmatic tocopherol induced antioxidant impact in caco-2 cells (abstract last paragraph). According to literature tocopherols have many antioxidative properties which seem to be independent of calcium homeostasis.
- In line with the last point why has only delta tocopherol a significant impact on the HyPer signal? According to the authors all tocopherols showed an increase in calcium level. As this is a major conclusion and result of this paper, the authors should give some experimental explanation and elucidate the underlying mechanisms more exactly.
Minor remarks:
Using a cell line over 20 passages (passage 20-40) might have a great impact on the cells and results. Especially when comparing the effect of different compounds more similar passages should be used.
Probably I have overseen it. What is control? Ethanol? What is the final solvent concentration?
Cytotoxicity should be measured after ionomycin, EGTA, etc. treatment in the used time schedule.
Reviewer 2 Report
The manuscript presents data which aims to link tocopherol with rapid intracellular free calcium transients in CaCo-2 cells. The results are potentially interesting, although there are several grammatical and methodological issues which the authors should, in this reviewer's opinion, respond to before the MS could be considered for publication.
With respect to grammar, the are a number of grammatical issues which, although minor, are distracting to the reader. For example, 'the' is frequently unnecessarily added to sentences, and in other cases it is absent where it is required. I will not mention all instances where there are these or other grammatical issues, but recommend the authors have the MS proof-read by an editor with greater familiarity with English language.
My greater concern is in the materials and methods, and with some of the interpretations by the authors. I elucidate these below
What type of coverslips were used? This affects not only cell growth, but also the ability to optimally use UV excitation; What was the percentage of DMSO that cells were exposed to for Fura-2 loading? This is critical to know, given the effect of this solvent on the plasma membrane; Why was the 340/380 ratio reported? CaCo-2 cells are relatively easy to calibrate intracellular free calcium values. It would have been preferable to see calibrated calcium concentrations, which would give some idea of the magnitude of calcium transients; When one chelates Ca++ with EGTA, the plasma membrane of cells can become destabilized. I would use caution in interpreting data so-obtained. It might have been more physiologically- sound to try cobalt substitutions, or better still the cell-impermeant form of BAPTA (tetrapotassium salt); Why were cells used at between 20-40 passages? this is quite a large range; On what basis were “responding cells” was a threshold of 3 standard deviations (SD) over baseline value used to consider an increase in cytosolic calcium? These findings are somewhat at odds with those of other cellular systems. For example, tocopherols prevent a rise in iron-induced cytosolic calcium in neurons (Crouzin et al., Volume 42, Issue 9, 1 May 2007, Pages 1326-1337). Could the authors comment on this? The MS I received did not have a clear indication of molar values; I suspect a formatting error caused the loss of milli- and micro- symbols; There was a large range in numbers of cells quantified in different experiments. What was the rationale?Author Response
Please see the attachment.

Reviewer 3 Report
This is an interesting article. However, I am very disappointed with the fact that there is not even ONE equation/formula to show the difference in the structure of tocopherols/tocotrienols and their reaction. What does actually take place when the measurements are being made? Unless some general introduction of the actual chemistry is presented this paper will be of interest to a small number of scientists who follow observations rather than actual mechanistic processes.
In addition – supplementary Fig 1/Fig. 2 figures are missing (maybe I missed them? In this case apologies are in place).
The Abstract is written in an unusual way- I have refereed other articles for Antioxidants and have never seen such an abstract that is subdivided in subdivisions (is this acceptable by the editorial board?).
Some English suggested revisions:
Line 13- which tocopherol? (the Greek letter is missing) Line 14- “offer a mixture of isomers? (please clarify) Line 47- change “after a lapse of 8h” to “and a lapse of 8h) Line 58-59: awkward sentence- rewrite please Line 64: place commas “γ-, as well as δ-tocopherols,… Line 65: replace “insult” with a more appropriate word Line71: replace “dependent of..” with “dependent on..” Line 71: replace “In parallel” with “Similarly” Line 72: replace “isomers” with “isomer’” Line 73: replace “than the other” with “any of the other” Material and methods: Lines 79-86: the word “acquired” is repeated too many times. Also where was absolute ethanol purchased from? Has the 30% hydrogen peroxide been standardized? Line 97: Where was pluronic acid purchased from?
Overall an interesting paper with a good error analysis.
Round 2
Reviewer 1 Report
The authors have already addressed several issues in their author´s reply and asked for more time to address experimentally some of my concerns.
I appreciate the willingness to perform these experiments and support the request of extension period. This is in principal an interesting paper and I would be glad if the authors could clarify the remaining points.
Author Response
Thanks again. We already bought the alpha-tocotrienol. We expect to have new data within a few days.

Reviewer 2 Report
Could the authors comment as to why they did not use calibrated intracellular calcium concentrations, instead of the ratiometric values
The HyPer biosensor, as the authors point out, responds primarily to H2O2. That said, does it really accurately reflect cellular redox status? Given the complex, diverse chemistry that is the sum of cellular redox potential, the best that can be said, in my opinion, is that HyPer reflects one component of cellular redox status - wording should be changed to reflect that.
Did the authors check the viability of CaCo-2 cells in the low Ca++ medium? This can be quite destabilizing on the cell membrane, and it would have been good to have tested this. If this was not done, I believe there should be a caveat to that effect
There are still some problematic grammatical issues. For example, on line 267, it states that. "... mediating changes in the antioxidant tone of cytoplasm." The meaning of the word 'tone', for example, is problematic.
In my copy of the PDF, there are no micro or milli symbols preceding the Molar units.
The authors speculate on intracellular sources of Ca++. Did they consider the GECKO series of Ca++ biosensors? These are genetically-engineered biosensors that can track nuclear vs mitochondrial vs ER release of Ca++
Round 3
Reviewer 2 Report
The MS is now acceptable